# Symptomatic Young Adults with ST-Segment Elevation—Acute Coronary Syndrome or Myocarditis: The Three-Factor Diagnostic Model

**DOI:** 10.3390/jcm11040916

**Published:** 2022-02-10

**Authors:** Paulina Wieczorkiewicz, Katarzyna Przybylak, Karolina Supel, Michal Kidawa, Marzenna Zielinska

**Affiliations:** Department of Interventional Cardiology, Medical University of Lodz, 92-213 Lodz, Poland; katarzyna.przybylak@umed.lodz.pl (K.P.); karolina.supel@umed.lodz.pl (K.S.); michal.kidawa@umed.lodz.pl (M.K.); marzenna.zielinska@umed.lodz.pl (M.Z.)

**Keywords:** myocarditis, ST-elevation myocardial infarction, young adults

## Abstract

Myocarditis may mimic myocardial infarction (MI) due to a similar clinical presentation, including chest pain, electrocardiography changes, and laboratory findings. The purpose of the study was to investigate the diagnostic value of clinical, laboratory, and electrocardiography characteristics of patients with acute coronary syndrome - like myocarditis and MI. We analysed 90 patients (≤45 years old) with an initial diagnosis of ST-segment elevation myocardial infarction; 40 patients (44.4%), through the use of cardiac magnetic resonance, were confirmed to have myocarditis, and 50 patients (55.6%) were diagnosed with MI. Patients with myocarditis were younger and had fewer cardiovascular risk factors than those with MI. The cutoff value distinguishing between myocarditis and MI was defined as the age of 36 years. The history of recent infections (82.5% vs. 6%) and C-reactive protein (CRP) levels on admission (Me 45.9 vs. 3.4) was more associated with myocarditis. Further, the QTc interval was inversely correlated with the echocardiographic ejection fraction in both groups but was significantly longer in patients with MI. Non-invasive diagnostics based on clinical features and laboratory findings are basic but still essential tools for differentiation between MI and myocarditis. The three-factor model including the criteria of age, abnormal CRP, and history of recent infections might become a valuable clinical indication.

## 1. Introduction

In spite of new diagnostic method developments, e.g., laboratory testing, echocardiography, and computed tomography angiography (CTA), the 12-lead electrocardiogram (ECG) still remains the pivotal and easily accessible tool for the patients with an initial diagnosis of acute coronary syndrome (ACS) in the emergency department (ED) [1]. According to the European Society of Cardiology (ESC) guidelines, reperfusion therapy should be implemented as soon as possible in patients with a suspicion of ST-segment elevation myocardial infarction (STEMI) [2]. However, especially among young patients with less typical symptoms, alternative causes of ST-segment elevation such as pericarditis, myocarditis, cardiomyopathy, pulmonary embolism, renal failure, aortic dissection, and toxin abuse may be considered [3,4]. Some previous reports reveal a prevalence of patients with suspected STEMI and non-obstructive coronary artery disease (CAD) up to 15% [4,5,6,7,8].

Acute myocarditis, due to its initial infarct-like clinical presentation, including chest pain, increased cardiac biomarkers, ST-segment elevation in the ECG, malignant arrhythmias, and impaired left ventricular ejection fraction (LVEF), constitutes a great challenge for clinicians to distinguish it from ACS [9,10]. The actual and accurate incidence of myocarditis in populations remains unknown, as the endomyocardial biopsy (EMB)—the diagnostic ESC gold standard—is infrequently followed in clinical practise [11,12]. It is estimated that myocarditis may show the global annual prevalence of approximately 22 people for every 100,000 [13]. It is still considered a poorly understood inflammatory disease of heterogeneous manifestation caused by a variety of factors, such as infections, systemic diseases, drugs, and toxins [11,14,15]. Although myocardial infarction (MI) and myocarditis are diseases of a diverse aetiology, differentiation between them in ED on the grounds of the clinical symptoms, ECG changes and laboratory markers is challenging. Thus, coronary angiography has still been performed on patients with inflammatory heart diseases in order to exclude ACS [16]. Taking into account possible adverse effects, few coronary risk factors in young patients, such as a history of fevers or recent viral illness (<3 weeks), the requirement for coronary angiography may be questionable [17]. Further investigations considering the preparation of a diagnostic algorithm based on non-invasive methods, including laboratory tests, echocardiography speckle tracking imaging, CTA, and cardiac magnetic resonance (CMR) parameters, are necessary to improve the rate of myocarditis detection when the EMB is not available or contraindicated [18]. Our study aimed to evaluate the role of clinical, laboratory, and ECG characteristics in young adults (≤45 years old) with ST-segment elevation in their ECG to set the diagnosis and estimate a prognosis of acute myocarditis vs. acute myocardial infarction (AMI). The following article was presented in accordance with the STROBE reporting checklist.

## 2. Materials and Methods

A total of 90 symptomatic patients (≤45 years old) with an initial diagnosis of STEMI were urgently admitted to the Department of Interventional Cardiology (Medical University of Lodz, Poland) within 12 h of onset symptoms, between April 2014 and December 2019, respectively. They were then analysed. All the data were collected from patients’ medical records and independently assessed by two investigators. ST-segment elevation was defined as a J-point elevation in two contiguous leads with the cut points: ≥0.25 mV in men below the age of 40 years, ≥0.2 mV in men over the age of 40 years, ≥0.15 mV in women in leads V2–V3, and/or ≥0.1 mV in other leads. In order to confirm or exclude obstructive CAD, the invasive coronary angiography was performed in all cases. Patients were classified into two groups, according to their final diagnosis. Group One: patients diagnosed with MI based on the clinical presentation, ECG recording, and the presence of a culprit lesion on the coronary angiography in accordance with the ESC Guidelines on Management of Acute Myocardial Infarction in Patients Presenting with ST Segment Elevation (2017). Group Two: patients with a normal coronary angiogram, diagnosed with acute myocarditis due to the clinical presentation and typical findings in CMR. CMR was performed during the same hospitalisation on a 1.5T scanner (Siemens Magnetom Avanto) with phased-array body coil, and ECG monitoring and enhanced contrast (Gadovist, Bayer Schering Pharma, Berlin, Germany) used dedicated syngo.via MR Cardiac Analysis software. Diagnosis of acute myocarditis was confirmed according to the updated 2018 Lake Louise criteria on the whole population [19,20]. According to the protocols recommended by the Society for Cardiovascular Magnetic Resonance, regional or global myocardial oedema and non-ischeamic myocardial injury were identified as areas of high-signal intensity in T2-weighted imaging and the regional late gadolinium enhancement (LGE) signal increase, respectively [21]. EMB was performed in no cases. Only seven symptomatic patients fulfilling the criteria of age and ST-segment elevation in their ECG on admission, but who were finally diagnosed with aortic dissection, toxin abuse, Takotsubo syndrome, pericarditis, or myocardial infarction type two, were excluded from the study. In the case of patients with toxin abuse or Takotsubo syndrome, the CMR was also performed.

During the course of patients’ examination, we analysed clinical presentation focusing on the chest pain, dyspnoea, and cardiac arrhythmias. Furthermore, we analysed the results of ECG, echocardiography, and CMR findings. With reference to the laboratory findings, we assessed the highly sensitive cardiac troponin T (TnT) level as a biomarker for the myocardial damage and the C-reactive protein (CRP) as a routine inflammatory indicator. The TnT cutoff point was 14 ng/L. The abnormal CRP level was defined as the value > 5 mg/L. All cardiovascular risk factors, including arterial hypertension, family history of CAD, smoking, diabetes mellitus, obesity defined as a body mass index (BMI) ≥ 30 kg/m^2^ [22], and other clinical data such as fever, infectious illness symptoms, or antibiotic therapy (<3 weeks), were also reviewed. The ECG parameters including QTc interval (corrected for heart rate with the Bazett formula) were manually analysed on admission. Echocardiographic examinations were performed during the first 48 h since admission using the Vivid E9 Ultrasound System with a 3.5 Mhz transducer (GE Healthcare, Horten, Norway) and analysed offline using the EchoPAC, Horten, Norway version 201 software.

The study was conducted in accordance with the Declaration of Helsinki, and the protocol was approved by the Local Ethics Committee (No RNN/03/20/KE).

All the statistical analyses were performed using the Statistica 13.1 (StatSoft Inc., Tulsa, OK, USA).

The classified variables were expressed as numbers and percentages, while the continuous variables were expressed as medians and interquartile ranges (IQR). The continuous values were analysed with the Mann–Whitney U test, whereas the chi-square test was used for the discrete values. Receiver operator characteristic (ROC) curves were constructed to determine the optimal cutoff values. In addition, the analysis was expanded to Spearman’s rank correlation coefficient. A *p* value of < 0.05 was considered to indicate a statistical significance.

## 3. Results

A total of 1384 patients were admitted to the ED with an initial diagnosis of STEMI between April 2014 and December 2019. A total of 90 consecutive patients ≤45 years old, including 14 females (15.6%) and 76 males (84.4%), were further enrolled. Three patients (3.3%) died during hospitalisation and all of them suffered from MI. The detailed distribution of patients according to their final diagnosis is presented in Figure 1.

In order to confirm or exclude obstructive CAD, the coronary angiography was performed on the whole population. Amid patients who had obstructive CAD excluded, the CMR examination based on updated 2018 Lake Louise criteria (increased signal intensity in T2-weighted imaging and increased signal intensity with a non-ischeamic distribution pattern in LGE images) confirmed the diagnosis of acute myocarditis in 40 cases. Pericardial effusion or pericardial signal abnormalities were not visualised in any case. The median number of left ventricle (LV) segments showing myocardial oedema on T2-weighted sequences was two (IQR 1–2). The minimum number of segments was one and the maximum was eight. Global oedema was not registered in any of the patients. The median number of LV segments showing LGE on the post-contrast sequence was four (IQR 4–6). The minimum number of segments was two and the maximum was eight (Table 1).

The baseline characteristics of the patients are presented in Table 2.

The median age was 26 (IQR 21.5–34.5) and 41 years (IQR 39–44) for myocarditis and MI, respectively (*p* < 0.001). The most common presenting symptoms were chest pain (94.4%, N = 85) and dyspnea (46.7%, N = 42). Fever occurred in 15 patients (16.7%) and the majority of them suffered from myocarditis (*p* < 0.001). All patients were reviewed in terms of cardiovascular risk factors, including current smoking, diabetes mellitus, arterial hypertension, obesity, and family history of CAD. In comparison to patients with myocarditis, a history of smoking, obesity, and arterial hypertension were more frequently found amid patients with MI (*p* < 0.05). There was no significant difference in the prevalence of diabetes mellitus and family history of CAD, whereas a history of recent infection of <3 weeks or antibiotic therapy within three weeks before admission was more associated with myocarditis diagnosis.

All patients were monitored in respect to, among other things, lipid profile, CRP, and TnT levels during the hospital follow-up. The lipid profile was abnormal in both groups; however, higher levels of total cholesterol (TC), low-density lipoprotein (LDL), and triglycerides (TG) were observed in MI patients (*p* < 0.001). There was no significant difference in the high-density lipoprotein (HDL) concentration between the groups. As for TnT, the concentration on admission and after 24 h were taken into account. The TnT patterns for both myocarditis and MI are presented in Figure 2.

Amid patients diagnosed with myocarditis, there was observed higher TnT levels on admission (Me 569.5 IQR 200–1074) with a relatively low peak in 24 h observation (Me 936 IQR 367–1364), in contrast to patients with MI. The median TnT level on admission for MI was 150 (IQR 55–437) with a higher peak after 24 h (Me 2088.5 IQR 757–4394). Both differences occurred to be statistically significant (*p* < 0.05).

The subsequent analysis has revealed the statistical significance between TnT after 24 h observation and LVEF in patients diagnosed with acute coronary syndrome (*p* < 0.05; R = −0.46), indicating that a low TnT peak value would be an essential marker of a better outcome. The corresponding relationship was not confirmed in the myocarditis group, or for TnT on admission in both.

As part of the initial assessment, the CRP level was also evaluated. The median value on admission was 45.9 (IQR 17.2–121) for patients with CMR-confirmed myocarditis and 3.4 (IQR 1.8–9.8) for patients with a final diagnosis of MI (*p* < 0.001), with the cutoff value of 16.1 for the acute myocarditis diagnosis. The area under the receiver operating characteristic (ROC) curve (AUC) was 0.869 (standard error = 0.042, 95% confidence interval: 0.786–0.952). There was also a statistically significant difference between the groups, considering CRP peak values during the observation (*p* = 0.014).

In the subsequent analysis, we also generated a ROC curve to test the sensitivity and specificity of the age for the diagnosis of acute myocarditis. The highest combination of sensitivity (90%) and specificity (77.5%) was obtained at the cutoff point for the age of 36 years. The AUC was 0.875 (standard error = 0.041, 95% confidence interval: 0.795–0.954) (Figure 3).

The detailed distribution of variables including age, the CRP value on admission, and the history of recent infection of <3 weeks, depending on the final diagnosis, is depicted in the 3D Figure 4a,b.

It graphically presents that patients suffering from acute myocarditis are younger, with abnormal levels of CRP values and past histories of recent infections, in contrast to patients with AMI. Taking into account all the three parameters collectively (history of recent infection, abnormal CRP value on admission >5 mg/L, and an age < 36 years), there are no patients in the MI group and there are as many as 27 patients (67.5%) diagnosed with myocarditis fulfilling the three criteria altogether (Figure 5).

Further, the ECG parameters with special regards to the QTc interval on admission were analysed. The median QTc duration for acute myocarditis was 413 ms (IQR 319–561) and 452 ms (IQR 366–621) for MI with *p* < 0.001. Statistical assessment has disclosed a significant correlation between the QTc interval duration and the LVEF in MI, myocarditis group, and both with respective values of *p* < 0.001, r = −0.5739; *p* < 0.05, r = −0.3389; *p* < 0.01, r = −0.5576. The cutoff point for increased risk of myocardial injury was 440 ms with the AUC of 0.735 (standard error = 0.055, 95% confidence interval: 0.627–0.844) (Figure 6).

During the course of the hospitalisation, all patients enrolled in our study had an echocardiography performed. No wall motion abnormality was detected in merely six patients with MI and as many as 21 patients with a final diagnosis of myocarditis. The median left ventricular ejection fraction (modified Simpson’s rule) was 58% (IQR 53–60) and 50% (IQR 45–50) for myocarditis and MI, respectively.

## 4. Discussion

Differentiation between myocarditis and MI remains one of the most crucial issues for the clinicians who meet young patients with ST-segment elevation in their ECG during their hospital practise. The clinical picture, including chest pain, dyspnea, and ECG changes, is not specific; hence, correct diagnostic and therapeutic processes still remain a great challenge in the ED. In our study, patients ≤ 45 years old constitute 6.5% of all admissions with an initial diagnosis of STEMI; 44.4% were finally diagnosed with acute myocarditis. The research might reveal that these patients are supposed to be younger, of male sex, and with a more frequent history of recent infection or antibiotic therapy [4,14,18]. On the other hand, obesity (defined as BMI > 30 kg/m^2^), smoking, and arterial hypertension prior to admission, as the prominent cardiovascular risk factors, were significantly more related to MI. Arterial hypertension was present in 44%, smoking in 74%, and obesity in 36% of MI cases. In contrast to the previous studies, diabetes and family history of CAD occurred to be statistically insignificant whereas hyperlipidemia was significantly more associated with STEMI [4,17,23].

In young patients presenting ST-segment elevation in the ED coronary angiography, the first exam performed to exclude obstructive CAD. At present, in most cases, this invasive technique is much more available than CTA or CMR imaging [9].

The CTA has an excellent ability to exclude significant CAD, and when patients show normal coronary arteries, no further testing is required. The studies prove that patients with low and intermediate pretest probability of CAD (estimated using the Duke Clinical Score, which includes chest pain, age, gender, and traditional risk factors) benefit the most from CTA, with a sensitivity and specificity for the test of up to 100% and 93%, respectively [24]. According to Hoffmann et al., the diagnostic accuracy of CTA in patients with acute chest pain and the initial diagnosis of ACS in the ED is estimated at 100% sensitivity and 84% specificity [25]. Additionally, CT (computed tomography) is increasingly enumerated beside MRI as an alternative imaging tool for estimation of myocardial oedema or myocardial fibrosis, particularly in those with contraindications to the latter. Several studies have demonstrated that, similarly to MRI, in CT imaging the late contrast enhancement may visualise the difference between the normal and damaged myocardium [26,27].

The ESC guidelines published in 2017 define proceedings relating to patients with ST-segment elevation. They directly state the requirement of immediate reperfusion strategy (maximum expected delay of <120 min. from initial diagnosis of STEMI to choose coronary angiography over fibrynolysis) in patients with ST-segment elevation, fulfilling the criteria for suggestive, ongoing coronary artery acute occlusion. Therefore, only after ruling out obstructive CAD is there room for consideration of differential diagnosis of STEMI and any other additional imaging studies, especially including echocardiography and CMR [2]. Likewise, in the Position Statement of the *ESC of Cardiology Working Group on Myocardial and Pericardial Diseases*, published in 2013, coronary angiography is enumerated beside EMB as a procedure recommended for consideration for all patients with clinically suspected myocarditis [11]. Thus, further studies are necessary to meet demand for the new criteria, to make the diagnosis of myocarditis more reliable and to potentially avoid the use of invasive and more costly diagnostics towards, for example, increased use of CTA in clinical practice.

CMR has become an integral part of the work-up of suspected myocarditis. In 2009, the Lake Louise criteria for diagnosis of myocarditis were based on traditional CMR techniques, including assessment of signal intensity in T2-weighted imaging and early and late gadolinium enhancement. It was suggested to assume a high likelihood of acute myocarditis if the CMR images indicated that two out of three criteria were fulfilled. In recent years, newly developed parametric mapping techniques such as T1 and T2 time and extracellular volume (ECV) have emerged as important complements to the previous components. These advances found their application in the 2018 update to the Lake Louise criteria, which recommend use of at least one T1-based sequence (T1 mapping and increase of ECV or LGE) and at least one T2-based technique (T2 mapping or T2-weighted sequence). The presence of a signal abnormality on both (“two out of two”) T1 and T2-based imaging constitutes a diagnosis [19,28]. It seems that introduction of the mapping techniques may increase the sensitivity of the acute myocarditis detection while maintaining its specificity [29]. However, it remains important to take into account that myocardial inflammation and myocardial injury are not specific to viral myocarditis, but they also occur in acute MI, cardiotoxity, and Takotsubo syndrome. The literature underlines the crucial role of the integration of clinical data, necrosis biomarkers, and imaging examinations in the whole diagnostic process [30,31].

A differential diagnosis between acute MI and myocarditis on the basis of laboratory findings may sometimes be difficult. Both diseases might be presented with increased levels of cardiac serum enzymes and inflammatory biomarkers [4,32,33]. However, a TnT level within normal limits does not exclude the diagnosis of myocarditis but may constitute a prognostic factor of a better outcome [27,34,35]. Serum cardiac biomarkers are routinely measured in patients presenting with chest pain in ED. Troponin is released into circulation from the damaged myocardium and is widely used as an indicator of the degree of myocardial injury [36]. In the case of myocarditis, cardiac serum enzymes are usually abnormal but there is an atypical release pattern. They are increased on admission with a suspected benign course and relatively late normalisation.

The subsequent parameters associated with diagnosis of myocarditis are electrocardiographic features. Beside ST-segment and T-wave changes, QRS duration, and QRS-T angle, the QTc interval is supposed to be associated with abnormal tissue characteristics and may play a role in risk stratification [37,38,39]. In our study, the inverse correlation between QTc duration and LVEF was revealed irrespective of the final diagnosis.

Based on our study, all of the above-mentioned features, including young age, absence of cardiovascular risk factors, history of fever or any other infection symptoms, atypical troponin release, and an increased CRP level on admission may compose a low-risk MI patient’s profile. Until now, there have been few studies sufficiently differentiating myocarditis and MI on the grounds of the non-invasive tests [9,13,17,33]. They have recommended assessment of the MI risk among young patients with ST-segment elevation before the invasive procedures. In our study, we attempted to find a cutoff value for the lowest risk of obstructive CAD. The highest combination of sensitivity and specificity of the test was obtained at the cutoff point for the age of 36 years. In our study, we suggest the three-parameter model, including the history of recent infection, an abnormal CRP value on admission > 5 mg/L, and the age of < 36 years, which might become a simple but essential tool for initial differentiation between acute myocarditis and MI in clinical practice.

Non-invasive diagnostics including clinical features, ECG patterns, abnormal laboratory findings, and the history of recent infection may constitute new starting points for consideration of the separation of the myocarditis group without additional imaging in patients for which the obstructive CAD had been excluded.

## 5. Limitations

Some limitations of the study need to be noted. Initially, this is a single-centre, retrospective study resulting in the relatively small number of patients. Secondly, despite the fact that EMB is still regarded as a gold standard for the diagnosis of myocarditis, according to the position of expert groups in ESC guidelines, it was not performed in any case. In our study, the diagnosis of acute myocarditis was confirmed in compliance with the updated 2018 Lake Louise criteria in the whole population (increased signal intensity in T2-weighted imaging and increased signal intensity with a non-ischeamic distribution pattern in LGE images). The novel techniques of myocardial CMR imaging (T1, T2 mapping, ECV) were not routinely performed.

## 6. Conclusions

In conclusion, in young patients with ST-segment elevation in their ECG, the differentiation is dominated by myocardial infarction and acute myocarditis. Due to their similar clinical presentation, this may sometimes prove difficult. Nevertheless, non-invasive diagnostics have the potential to become essential tools for the assessment of patients with suspected myocarditis. Taking into account the necessity of the obstructive CAD exclusion, it appears that for young patients in their twenties with chest pain in the ED, as well as a history of recent infection and increased laboratory biomarkers of inflammation, the non-invasive testing for CAD—with particular attention to CTA—might be considered if it is fully available. For further optimisation of the diagnostic algorithms, multi-centre studies are required.

## Figures and Tables

**Figure 1 jcm-11-00916-f001:**
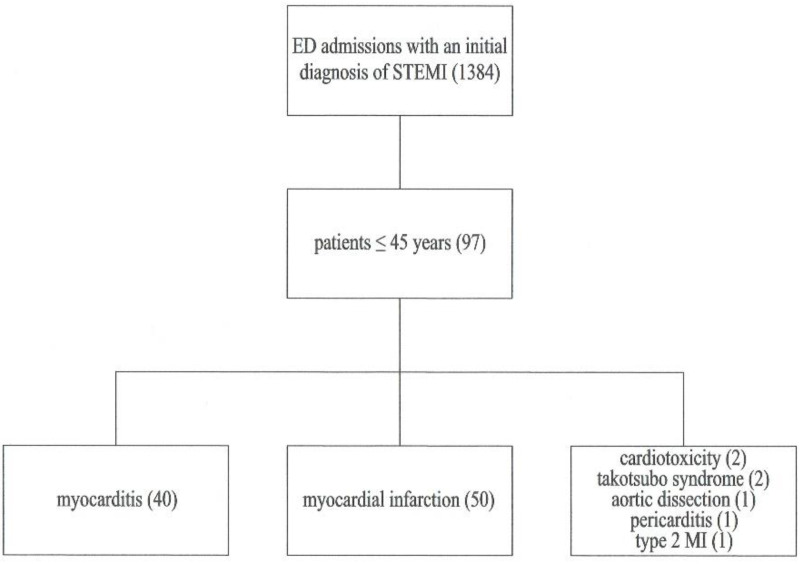
Study flow chart, MI—Myocardial infarction.

**Figure 2 jcm-11-00916-f002:**
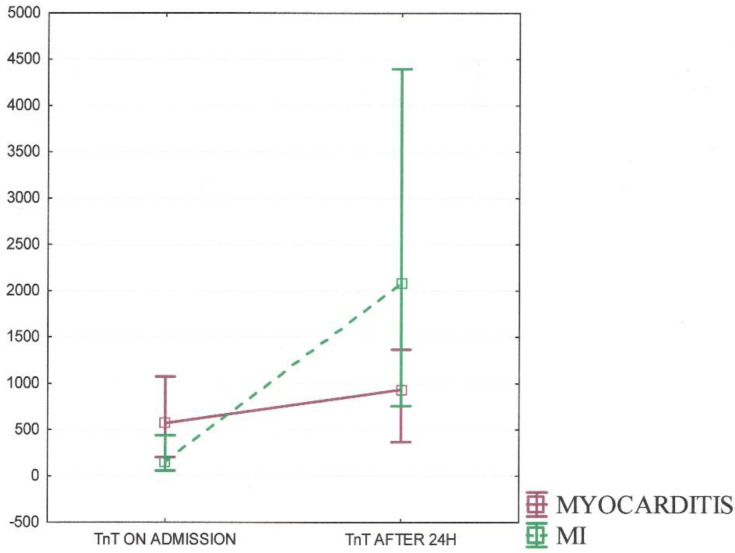
TnT patterns for myocarditis and myocardial infarction including TnT levels on admission and after 24 h.

**Figure 3 jcm-11-00916-f003:**
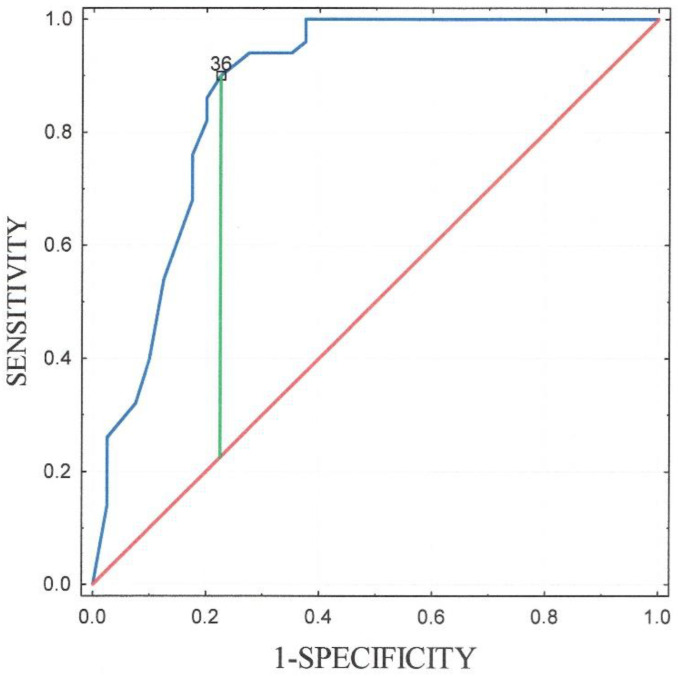
ROC curve of the age for the diagnosis of acute myocarditis. The AUC was 0.875 (standard error = 0.041, 95% confidence interval: 0.795–0.954). It shows that the sensitivity (90%) and specificity (77.5%) of the age is adequate in statistics.

**Figure 4 jcm-11-00916-f004:**
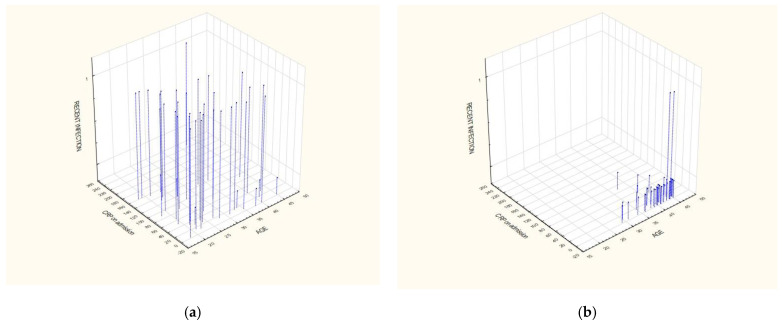
Three-dimensional charts showing the distribution of variables based on age, CRP level on admission, and history of recent infection: (**a**) Myocarditis; (**b**) Myocardial infarction.

**Figure 5 jcm-11-00916-f005:**
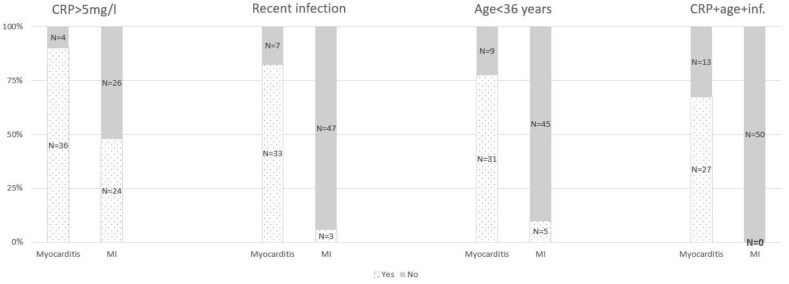
Multiparametric model including abnormal CRP level on admission > 5 mg/L, history of recent infection, and age < 36 years; all three are divided into groups presented as number of patients (%).

**Figure 6 jcm-11-00916-f006:**
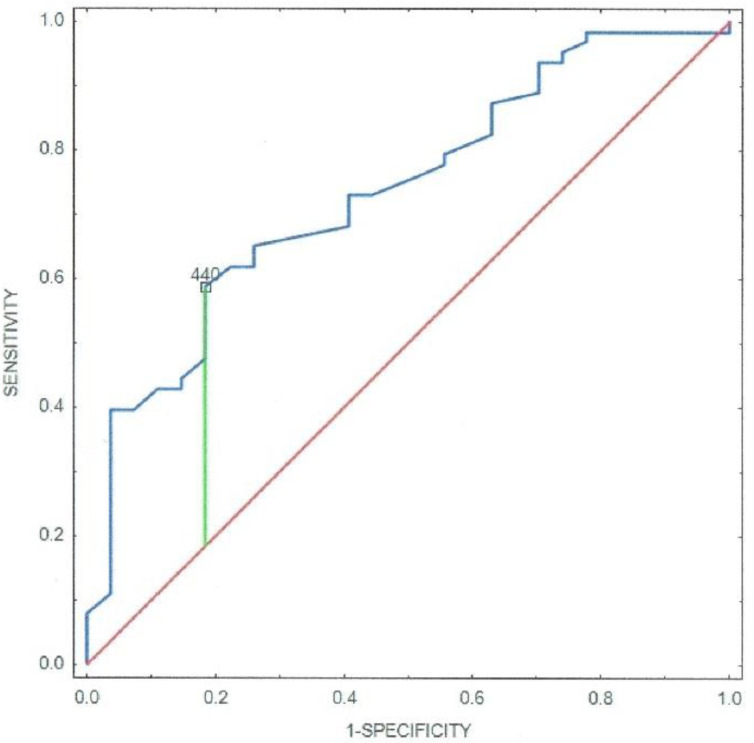
ROC curve of the QTc on admission for the ejection fraction measured during the first 48 h since admission. The AUC was 0.735 (standard error = 0.055, 95% confidence interval: 0.627–0.844).

**Table 1 jcm-11-00916-t001:** Cardiac magnetic resonance findings.

	Myocarditis N = 40
Ejection fraction (%)	57 (52–59)
LV segments with LGE	4 (4–6)
LV segments with oedema	2 (1–2)
Regional increase of T2 signal intensity *	40 (100%)
Global T2 signal intensity ratio ≥ 2 *	0
LGE non-ischeamic pattern *	40 (100%)
Pericardial effusion/pericardial abnormalities	0

All values presented as median (IQR) or N (%). * Inclusion criteria.

**Table 2 jcm-11-00916-t002:** Baseline characteristics.

	Myocarditis N = 40	Myocardial Infarction N = 50	*p* Value
Age	26 (21.5–34.5)	41 (39–44)	<0.001
Sex (male)	37 (92.5%)	39 (78%)	0.593
Smoking	16 (40%)	37 (74%)	<0.001
Hypertension	4 (10%)	22 (44%)	<0.001
Diabetes	2 (5%)	4 (8%)	0.592
Obesity BMI > 30 (kg/m^2^)	5 (12.5%)	18 (36%)	0.011
Family history of CAD	32 (80%)	36 (72%)	0.331
Symptoms			
Chest pain	38 (95%)	47 (94%)	0.837
Dyspnea	3 (7.5%)	39 (78%)	0.059
Fever *	14 (35%)	1 (2%)	<0.001
Recent infection *	33 (82.5%)	3 (6%)	<0.001
Antibiotic therapy *	9 (22.5%)	1 (2%)	0.002
Left ventricular ejection fraction (%)	58 (53–60)	50 (45–55)	<0.001
Regional wall motion abnormality	19 (47.5%)	44 (88%)	<0.001
Hospital mortality	0	3 (6%)	0.115
Laboratory findings			
TnT (on admission) (ng/L)	569.5 (200–1074)	150 (55–437)	0.025
TnT (after 24 h) (ng/L)	936 (367–1364)	2088.5 (757–4394)	<0.001
CRP (on admission) (mg/L)	45.9 (17.2–121)	3.4 (1.8–9.8)	<0.001
CRP (max.) (mg/L)	47.9 (17.8–122.3)	15.9 (2.2–89.7)	0.014
Total cholesterol (TC) (mmol/L)	3.89 (3.6–4.66)	5.95 (4.84–6.79)	<0.001
LDL	2.4 (1.83–2.86)	3.44 (2.69–4.55)	<0.001
HDL	1.1 (0.87–1.29)	1.1 (0.96–1.39)	0.310
Triglycerides (TG)	1.19 (0.73–1.6)	1.815 (1.34–2.59)	<0.001
ECG parameters			
QTc (ms)	413 (319–561)	452 (366–621)	<0.001

All values presented as median (IQR) or N (%). * <3 weeks.

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
