# Peer review of "Symptomatic Young Adults with ST-Segment Elevation—Acute Coronary Syndrome or Myocarditis: The Three-Factor Diagnostic Model"

_jcm, 2022, doi:10.3390/jcm11040916_

Round 1

Reviewer 1 Report

The study compared a group of young individuals with clinically-suspected acute myocarditis (diagnosed by CMR + no coronary artery disease) versus a group of STEMI patients. Despite the difference in mean age (as much as 15 years) and risk factors for coronary artery disease, abnormal level of CRP value and past history of recent infection favored myocarditis diagnosis. Even if  differential diagnosis between these two conditions is certainly a clinical issue. I believe the findings of the present study are largely confirmatory. One other comment: the ECG analysis was mainly based on QTC values but lacked important information such as the prevalence of PR-segment depression and the amplitude and distribution of ST-segment elevation

Author Response

Dear Sir/Madam,

On behalf of all co-authors, I would like to thank you very much for your all comments and suggestions.

In our study, as far as the electrocardiographic analysis is concerned, we have focused mainly on QTc values. While collecting data we have taken into account more ECG parameters but finally, they have occurred to be stastistically insignificant.

Among our patients there was only one who presented the prevalence of PR segment depression.

According to ESC guidelines ST segment elevation was defined as a J point elevation in two contiguous leads with the cut-points: ≥0.25 mV in men below the age of 40 years, ≥0.2 mV in men over the age of 40 years, ≥0.15 mV in women in leads V2 – V3 and/or ≥0.1 mV in other leads. When patients fulfilling the above-mentioned criteria, the coronary angiography was performed. Because it would not change the procedures on Emergency Department and not have an influence on the inclusion criteria, we have not analysed the amplitude of ST segment elevation.

Hopefully, the distribution of ST segment elevation will be the subject of our further study. Among others we are going to analyse the ECG changes in comparison to MRI and echocardiography findings.

Best regards,

Paulina Wieczorkiewicz

Reviewer 2 Report

Dear Authors,

It was a pleasure to review the article “Symptomatic young adult with ST segment elevation – acute coronary syndrome or myocarditis?”. The authors retrospectively studied 90 young patients with <45 years old, admitted with ST-elevation on ECG that were diagnosed with ST-elevation myocardial infarction (STEMI) or myocarditis, and evaluated the role of clinical, laboratory and ECG characteristics that could help in the differentiation between these two pathologies. They showed that patients with myocarditis, compared to those with myocardial infarction, were younger; had a lower prevalence of cardiovascular risk factors, such as smoking, hypertension and obesity; had a higher prevalence of fever at admission, history of recent infection or antibiotic therapy;  higher left ventricular ejection fraction (LVEF) and less regional wall motion abnormalities; higher Troponin T level on admission but lower peak in 24 hours; higher C-reactive protein (CRP) at admission and peak during hospital stay; lower total cholesterol, LDL and triglycerides levels; and lower QTc interval. Furthermore, the authors created a model considering three parameters: history of recent infection, abnormal CRP value on admission >5 mg/l and the age<36 years, and showed that in the group of patients fulfilling the three criteria there were no cases of myocardial infarction. Finally, the authors showed an inverse correlation between QTc interval and LFEF.

The 3-variable score is very interesting and is of potential clinical utility in the future. I have few comments presented below:

  1. The main utility of this work is the 3-parameters model created to distinguish between STEMI and myocarditis, and I think it should be more highlighted in the title, in the abstract and in the discussion.
  2. The duration of symptoms in patients presenting with STEMI or myocarditis should be outlined, as this can influence the values of troponin and CRP at admission.
  3. In the discussion, besides commenting the cardiac CT role to exclude coronary artery disease, the authors should comment its potential role to identify edema/inflammation to help confirming the diagnosis of acute myocarditis.
  4. There should be mentioned if other ECG parameters, such as the extension of ST elevation, was helpful to distinguish between STEMI and myocarditis.
  5. “Based on our study, all the above-mentioned features including: young age, male sex,…” Male sex was not a statistically significant parameter in the distinction between STEMI and myocarditis.
  6. A limitation paragraph is needed

Author Response

Dear Sir/Madam,

On behalf of all co-authors, I would like to thank you very much for your all comments and suggestions.

According to your indications we have underlined the three – parameter diagnostic model including history of recent infection, abnormal CRP value on admission >5mg/l and the age <36 years between the groups. We have mentioned it additionally in the title, abstract and discussion section.

All of our patients were urgently admitted within 12h of symptom onset. Simultaneously, with respect to the ESC guidelines on management of acute myocardial infarction in patients presenting with ST-segment elevation, it was a time criterion for performing an emergency coronary angiography. We have added this information in the materials and methods section.

Following your suggestions, we have added the paragraph concerning the potential role of CT to estimate the myocardial damage. It was not only the great diagnostic tool to confirm the diagnosis of obstructive coronary artery disease, aortic dissection and pulmonary embolism, but also to estimate the left ventricular ejection fraction and visualize the tissue abnormality particularly in those with contraindication to MRI.

In our study, as far as the electrocardiographic analysis is concerned, we have focused mainly on QTc values. While collecting data we have taken into account more ECG parameters but finally, they have occurred to be stastistically insignificant. According to ESC guidelines when patients fulfilling the criteria of ST segment elevation, the coronary angiography was performed. Because it would not change the procedures on Emergency Department and not have an influence on the inclusion criteria, we have not analysed the amplitude of ST segment elevation.

The wording ‘’male sex’’ was removed from the manuscript.

The limitation paragraph was added.

Best regards,

Paulina Wieczorkiewicz

Reviewer 3 Report

A brief summary

The article is aimed to evaluate the role of clinical, laboratory and ECG characteristics in young adults (≤45 years old) with ST segment elevation in ECG to set the diagnosis and estimate a prognosis of acute myocarditis vs. acute myocardial infarction (AMI). Tne topic is up to date for studing because of difficalties of miocarditis and AMI dizgnostics in time among young adults. I fully agree with authors that the noninvasive diagnostics based on the clinical features and laboratory findings is basic but still essential tool for differentiation between MI and myocarditis.

General concept comments

Please:

  1. add the citting for the criteria of obesity (page 2 - «obesity defined as a body mass index (BMI) >30kg/m2» and explain why the BMI is >30kg/m2, but not ≥30kg/m2;
  2. correct the missprint on the page 4 «The most common presenting symptoms were chest pain (94.4%, N=95 – N Should be 85);
  3. add the exact sensitivity and specificity value of the age of 36 years - page 5 «The highest combination of 172 sensitivity and specificity was obtained at the cut-off point for the age of 36 years» and figure 3.

 The article completly covers the topic, is presented in a well-structured manner, the references are current and appropriate (mostly within the last 5 years – 55%), the figures/tables properly show the data, the conclusions are consistent with the results.

Author Response

Dear Sir/Madam,

On behalf of all co-authors, I would like to thank you very much for your all comments and suggestions.

According to your indications, we have added the missing value of the exact sensitivity and specificity of the age of 36 years. This is 90% and 77.5% respectively. We have also put these values ​​on the image caption.

We have corrected the missprint on the page 4. It is N=85. We apologize for this mistake.

As far as the criteria of obesity are concerned, we based on the European Guidelines for Obesity Management in Adults [21] which define obesity as a body mass index ≥30kg/m2. It is our missprint too. We have corrected the interval symbol.

Best regards,

Paulina Wieczorkiewicz

Round 2

Reviewer 1 Report

Thank-you for your reply

Author Response

Dear Sir/Madam,

I would like to thank you once again for your all comments and suggestions.

Best regards, 

Paulina Wieczorkiewicz